ecology, environmental science

ecosystem, myctophid, mesopelagic fish, acoustics, biomass, Southern Ocean

**Author for correspondence:**
Tracey Dornan
e-mail: td15166@bristol.ac.uk

### PUBLISHING

# Swimbladder morphology masks Southern Ocean mesopelagic fish biomass

Tracey Dornan[1,2], Sophie Fielding[1], Ryan A. Saunders[1] and Martin J. Genner[2]

[1]British Antarctic Survey, High Cross, Madingley Road, Cambridge CB3 0ET, UK
[2]School of Biological Sciences, University of Bristol, Life Sciences Building, 24 Tyndall Avenue, Bristol BS8 1TQ, UK

TD, 0000-0001-8265-286X

Within the twilight of the oceanic mesopelagic realm, 200–1000 m below sea level, are potentially vast resources of fish. Collectively, these mesopelagic fishes are the most abundant vertebrates on Earth, and this global fish community plays a vital role in the function of oceanic ecosystems. The biomass of these fishes has recently been estimated using acoustic survey methods, which rely on echosounder-generated signals being reflected from gas-filled swimbladders and detected by transducers on vessels. Here, we use X-ray computed tomography scans to demonstrate that several of the most abundant species of mesopelagic fish in the Southern Ocean lack gas-filled swimbladders. We also show using catch data from survey trawls that the fish community switches from fish possessing gas-filled swimbladders to those lacking swimbladders as latitude increases towards the Antarctic continent. Thus, the acoustic surveys that repeatedly show a decrease in mesopelagic fish biomass towards polar environments systematically overlook a large proportion of fish species that dominate polar seas. Importantly, this includes lanternfish species that are key prey items for top predators in the region, including king penguins and elephant seals. This latitudinal community switch, from gas to non-gas dominance, has considerable implications for acoustic biomass estimation, ecosystem modelling and long-term monitoring of species at risk from climate change and potential exploitation.

## 1. Introduction

Mesopelagic fish inhabit the twilight zone of the world's oceans, 200–1000 m below sea level. This global community of typically small (less than 20 cm) fish is often dominated by myctophids, commonly known as lanternfishes (Family Myctophidae) by both abundance and biomass [1]. Debate surrounds the magnitude of mesopelagic fish biomass, with global estimates ranging from 1 to 19.5 gigatonnes [1–3]. A key issue underlying this uncertainty is that many mesopelagic fish, including lanternfishes, exhibit net avoidance behaviour, potentially resulting in an underestimation of biomass [4].

Active acoustics provides a more informative method of studying these animals at the oceanic scale. Acoustic surveys are routinely used to estimate the biomass of commercially important fish stocks [5]. The underlying principle of active acoustics is to transmit a pulse of sound of known frequency and duration into the water column from an echosounder; when the sound-wave encounters something of a different acoustic impedance, such as gas in the swimbladder of a fish, it is reflected or scattered back to the transducer. The quantity of reflected signal or 'echo' is then integrated throughout the water column, and is commonly used as a proxy for biomass [2,6]. However, the interpretation of acoustic data into meaningful biology is complex, and requires ancillary information on species distribution, behaviour and fish

morphology [7], as well as knowledge of how a specific target organism backscatters the acoustic signal at a given acoustic frequency [6].

Gas in the swimbladders of fish can account for up to 95% of reflected acoustic 'backscatter' signal [8]; thus the swimbladder morphology of fish is critical for determining the effectiveness of active acoustics for estimating fish biomass. It has been known for over 50 years that mesopelagic fishes can differ in swimbladder morphology [9], with species showing both intra- and interspecific variability. For example, some species can maintain a gas-filled swimbladder throughout their lifespan, while some species may never have a gas-filled swimbladder, and others lose the gas component in adulthood [9]. Net sampling is regularly used to groundtruth acoustic data, providing knowledge of the species present and their morphological characteristics [10]. However, this is challenging to undertake comprehensively at the ocean basin scale [11] and adequate net sampling has generally focused on commercially harvested species at smaller regional scales.

In the Southern Ocean, 35 species of myctophids are known to occur [12], where they form a key component of the Antarctic ecosystem, acting as both predators of zooplankton [13–15] and prey for higher predators, including seabirds and seals [16–19]. In this food web, which is typically dominated by krill (*Euphausia superba*), myctophids have elevated importance for higher-trophic-level species during the years when krill are scarce [20]. Additionally, these myctophid species play a key role in carbon transport through the water column during diel vertical migration (DVM), which may contribute up to 17% of total carbon export from the system [21]. Assessment of the biomass of these species is important for our understanding of ecosystem function and carbon sequestration, both regionally and globally. However, the utility of active acoustics for this assessment has been hampered by limited data on swimbladder morphology both within and among key myctophid species. Specifically, it has been unclear if the reported latitudinal decline in backscatter towards the Antarctic continent [22,23] is a consequence of a decrease in fish biomass, or instead a consequence of the coincidental change in mesopelagic fish community composition [23].

Here, we report a detailed exploration of the potential influence of swimbladder morphology on estimates of mesopelagic fish biomass in the Southern Ocean, which for the purposes of this study we define as the region south of 50° S. We first use multiple acoustic transects to confirm a pattern of declining acoustic backscatter towards the Antarctic landmass in the South Atlantic, in agreement with observations from the South Pacific sector [23]. We then analyse the swimbladder condition of the common myctophid species in the region using X-ray imaging of fresh specimens, dissection of fresh specimens and X-ray micro-computed tomography (CT) of preserved specimens. Finally, we use net data to describe the change in the mesopelagic community towards higher latitudes. We conclude that the reduction in backscatter with latitude towards Antarctica is strongly influenced by a shift in community structure from gas-bladdered to non-gas bladdered species. We consider this result from the perspective of acoustic biomass assessment, and discuss the potential underlying ecological and evolutionary drivers of the observed shift in myctophid community composition and morphology.

## 2. Methods

### (a) Acoustic surveys

We quantified nautical area scattering coefficient (NASC, $m^2$ $nmi^{-2}$), a measure of mean water column acoustic backscatter and a proxy for biomass, in relation to latitude. Six acoustic transects from five individual cruises between the Falkland Islands and the South Orkneys were conducted aboard the RRS *James Clark Ross*, covering Austral spring to autumn (figure 1). An EK60 split-beam hull-mounted transducer was used to collect 38 kHz data to depths of 1000 m on all cruises with the exception of JR161 and JR200, where data were collected to 800 m and 990 m, respectively. All data were calibrated, processed and integrated in 1 km distance by 10 m depth bins in ECHOVIEW (v. 8.0.95, Echoview Software Pty Ltd, Hobart, Australia). Prior to integration, bad or unwanted data such as false bottom echoes, seabed, surface near-field, intermittent noise and attenuated signal were set to 'no-data' and excluded from the analyses. Non-transit data, where vessel speed slowed below 4 knots to undertake alternative science operations, were not included in the analysis. After integration, data collected in water shallower than 1000 m were excluded from analysis to constrain the study to mesopelagic waters. Total water column NASC was calculated in R (v. 3.5.1) [27] and $log_e$-transformed prior to fitting a linear regression model using latitude as a predictor variable. To verify that high NASC values were valid and not noise, the top 1% of NASC values were visually scrutinized on echograms. Less than 10% of these were suspected to be noise-biased, and the biased NASC values were removed from further analysis. Both day and night collected acoustic data were used in the analysis. To confirm that DVM did not introduce bias, linear regressions were also carried out on separate day and night data, and all reported trends remained consistent (electronic supplementary material, figure S1).

### (b) Net sampling

Stratified net sampling was undertaken on six cruises, between 2004 and 2017, at locations spanning the major frontal positions and water masses of the Scotia Sea (figure 1). Nets were deployed day and night during early cruises (JR161 and JR177). These were later restricted to night only sampling (JR200, JR15004 and JR16003) due to comparatively low fish abundance within daylight catches, presumably due to net avoidance behaviour.

Samples were collected using an opening and closing rectangular mid-water trawl net system (RMT25) [28]. The RMT25 is equipped with two nets, with an aperture of 25 $m^2$, and cod-end mesh of 5 mm. To sample the mesopelagic and epipelagic regions, each haul was stratified into four depth zones: 1000–700 m, 700–400 m, 400–200 m and 200 m–surface. Nets were towed obliquely in each zone at a towing speed of approximately 2.5 knots, for a duration of 30–60 min. All nets were closed during deployment and recovery, to minimize contamination from different depth zones. Once on deck, cod-end samples were transferred to fresh seawater. The total catch weight of all fauna by species was recorded whenever possible. Fish were then placed on ice for identification, and the standard length (SL) measured, before either further morphological analysis on board the research vessel or preservation by freezing at −20°C.

Fish from these surveys were used for soft tissue X-ray and/or dissection (freshly caught specimens), or X-ray CT (frozen specimens). Additional fish for morphological analysis were sampled opportunistically from RMT8 and multiple opening/closing net and environmental sensing system (MOCNESS) nets deployed during the same cruises (electronic supplementary material, table S1).

### (c) Swimbladder gas assessment

The swimbladders of seven of the eight most common species of myctophid (based on the net data) were assessed for the presence

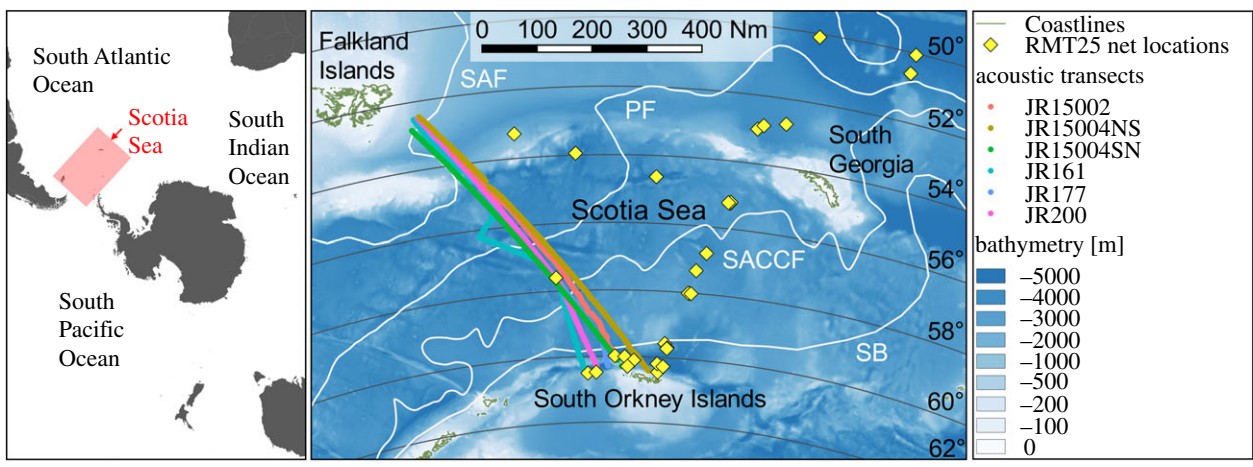

**Figure 1.** Study location in the Scotia Sea, Atlantic sector of the Southern Ocean. RMT25 surface to 1000 m depth net sample locations (yellow diamond). Acoustic transects between the Falkland Islands and the South Orkney Islands (coloured lines): spring cruises, JR161 (October 2006), JR15002 (November 2015); summer cruises JR177 (January 2008), JR15004 (January and February 2016); autumn cruise, JR200 (March 2009). Mean frontal positions are represented in white. SAF, Sub-Antarctic Front; PF, Polar Front; SACCF, Southern Antarctic Circumpolar Current Front; SB, Southern ACC Boundary [24–26]. Also shown are the 2° latitudinal bands used in analysis. Map generated in Quantum GIS v. 2.18 (www.qgis.org). (Online version in colour.)

or absence of gas: *Electrona antarctica* ($n = 56$), *Electrona carlsbergi* ($n = 28$), *Gymnoscopelus braueri* ($n = 21$), *Gymnoscopelus fraseri* ($n = 12$), *Gymnoscopelus nicholsi* ($n = 14$), *Krefftichthys anderssoni* ($n = 39$) and *Protomyctophum bolini* ($n = 32$). Assessment of individual fish was conducted using one of three methods: (a) visual inspection following dissection, (b) soft tissue X-ray scanning and (c) X-ray CT scans.

For visual inspection following dissection, freshly captured samples were dissected and the swimbladder punctured under water to record presence or absence of swimbladder gas. All dissections occurred within 8 h of capture, with fish stored in individual sealed bags at approximately 4°C prior to dissection.

All soft tissue X-ray images were captured using an Ultra-power 100 veterinary X-ray unit. Lateral and dorsal X-rays were taken with the film cassette positioned 0.88 m from the radiography unit. Exposure time and peak voltage (kVp) were set depending on the size and thickness of the animals being imaged, from small species being exposed for 0.08 s at 44 kVp, to larger species exposed for 0.09 s at 50 kVp.

Fish subjected to X-ray CT were scanned using one of two methods: (a) fish were freshly defrosted, held on ice in the CT facility, and mounted in polyethylene and foam to minimize movement in the scanner; and (b) fish were fixed in 5% formalin, stained with Potassium Iodide IKI, rinsed and scanned in distilled water; using a Nikon XTH225ST CT scanner. Fish were scanned in batches or individually depending on the size of the fish; and settings were adjusted between scans to capture the maximum detail while retaining all of the fish in view.

Swimbladders were considered to be gas-filled if they were found to contain gas or if the swimbladder was visibly ruptured on X-ray CT images, soft tissue X-ray images or during dissection. Fish were classed as non-gas-filled if they did not contain gas, or when gas was only present in the oesophagus/gut, indicative of ingestion of gas on hauling. Damaged fish, or those for which CT images were inconclusive, were excluded from analysis. Electronic supplementary material, tables S2 and S3 have detailed information on how gas presence or absence was determined from X-ray CT images.

Species not assessed for gas component as part of this study were assigned swimbladder status from the literature. *Protomyctophum tenisoni* was assigned as gas bearing, based on previously published analyses [9]. Non-myctophid Bathylagidae [29] and *Notolepis* spp. [30] do not possess swimbladders and so were not assessed for gas. As *Cyclothone* species were only identified

to genus level, all were treated as 'fat invested' (for justification see electronic supplementary material, table S4).

## (d) Statistical analysis

Community composition was determined from only the night-sampled, surface–1000 m depth stratified, RMT25 net samples, which were standardized for tow speed and duration. Analyses focussed on 11 of the most dominant Scotia Sea fishes, which accounted for greater than 94% of all fish captured by abundance in RMT25 net data (electronic supplementary material, table S4). A depth-integrated abundance of each species was assessed for each sampling event, by calculating the average abundance across the four depth zones. Latitudinal community change was assessed by calculating mean species abundance in 2° latitudinal bands. Fish biomass for each of the 11 fish species was derived directly from the same net samples as the abundance data. Where catch weights were missing, abundance of each species was multiplied by a mean weight for each species (calculated from combined JR161, JR177 and JR200 data). Swimbladder gas status was assigned, from either this study or literature as described above, to each individual in the net based on species, and SL where relevant. All statistical analyses were conducted in R (v. 3.5.1) [27].

## 3. Results

### (a) Acoustic backscatter declines with latitude

Significant declines in $\log_e$ NASC with increasing latitude were evident in all six acoustic transects (figures 1 and 2). The transect with the greatest variability along the linearly decreasing trend was undertaken during the late Austral spring cruise JR15002 (figure 2), where visual inspection of echograms revealed high, patchy levels of backscatter in the upper water column. To confirm that the declining trend in NASC was not associated with a decreasing biomass in general, the total biomass of all fauna (both fish and invertebrates) and fish (study species only), captured in each stratified net sample, standardized for tow speed and duration, were plotted against latitude. This revealed that there was no decrease in biomass with increasing latitude in net samples (electronic supplementary material, figure S2 and S3).

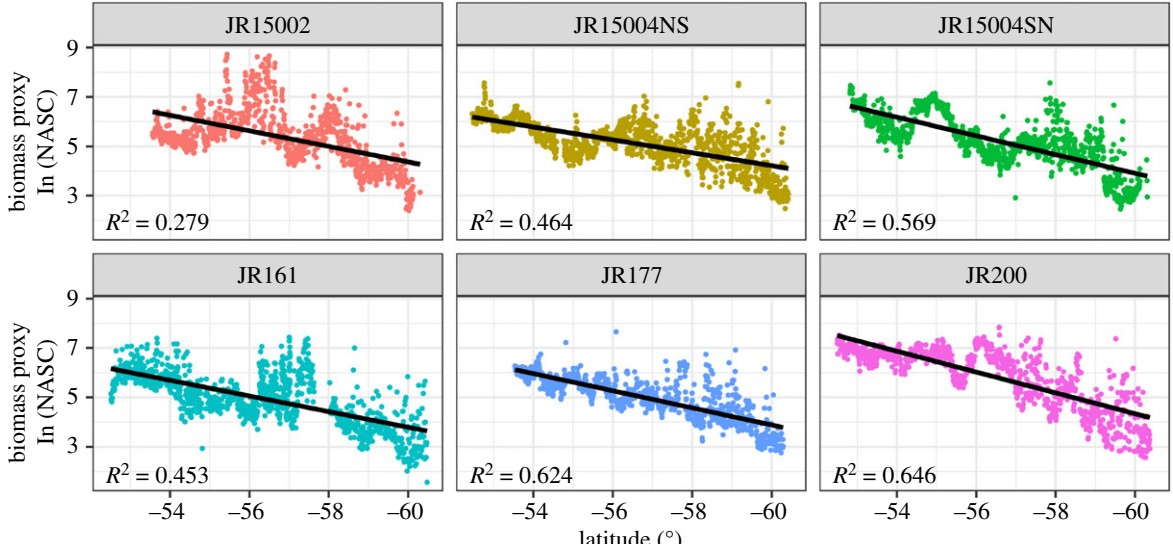

**Figure 2.** Relationship between the nautical area scattering coefficient (NASC, $m^2$ $nmi^{-2}$), a proxy for biomass, and increasing latitude by cruise number. JR15004 had both north to south (NS) and south to north (SN) transits, all others are one way only. All data shown were collected in water greater than 1000 m depth. Linear regressions (black lines) are statistically significant ($p < 0.001$). (Online version in colour.)

### (b) Gas presence and absence of key mesopelagic fish species

*Electrona carlsbergi* (SL 70–86 mm, $n = 28$), *K. anderssoni* (SL 30–70 mm, $n = 39$) and *P. bolini* (SL 29–62 mm, $n = 32$) all showed evidence of gas-filled swimbladders across all lengths assessed, indicative of gas presence throughout their lifespans. *Gymnoscopelus braueri* (SL 68–123 mm, $n = 21$), *G. nicholsi* (SL 124–153 mm, $n = 14$) and *G. fraseri* (SL 55–84 mm, $n = 12$) showed no evidence of swimbladder gas.

There was an apparent ontogenetic loss of swimbladder gas in *E. antarctica* (SL 27–103 mm, $n = 56$), with SL a highly significant predictor of the presence of gas ($p < 0.001$), and the modelled shift in probability of gas presence to absence estimated at SL 51.4 mm (electronic supplementary material, figure S4). Both dissection and X-ray CT images (figure 3a) revealed the swimbladder tissue to be thickened in larger specimens with no gas retained.

The swimbladder of *K. anderssoni* was thick-walled and possessed a fine transparent membraned oval structure at the anterior side, which was commonly inflated with a bubble-like appearance on dissected and CT-scanned specimens (figure 3b). Swimbladders of *E. carlsbergi* and *P. bolini* were apparently thin walled as they were commonly ruptured on hauling with gas filling abdominal cavity.

### (c) Changing community structure

The mesopelagic fish community was dominated by Myctophidae by abundance, accounting for 75.07% of fishes captured with the RMT25, with Bathylagidae and Gonostomatidae accounting for 14.41% and 6.30%, respectively. The eleven most commonly occurring mesopelagic taxa were selected for community assessment accounted for greater than 94% of individuals captured (electronic supplementary material, table S4). There was an overall reduction in species richness of mesopelagic fishes with increasing latitude, and a switch in the dominant species from the gas-bearing *P. bolini* and *K. anderssoni* at lower latitudes, to the regressed and non-gas-bearing swimbladder *E. antarctica* and *G. braueri* at higher latitudes (figure 3c).

### (d) Effects of changing community on acoustic signal—less backscatter, not fewer fish

Mean fish abundance (mean 0.867 individuals 1000 $m^{-3}$, range 0.751–0.920 individuals 1000 $m^{-3}$) and biomass (median 3.993 g 1000 $m^{-3}$, range 1.520–5.922 g 1000 $m^{-3}$) as estimated using RMT25 trawl samples were consistent across the latitudinal gradient of the Scotia Sea (figure 4a,b). To examine change in morphology with latitude all *Gymnoscopelus* species, *E. antarctica* > 51.4 mm, Bathylagidae [29] and *Notolepis* spp. [30] were assigned 'no gas' status. All *E. carlsbergi*, *P. bolini*, *K. anderssoni*, *E. antarctica* < 51.4 mm and *P. tenisoni* [9] were assigned as 'gas'. Cyclothone were assigned as 'fat invested'. This categorization revealed a clear latitudinal shift in the community from strongly scattering gas-bladdered species in the north of the sampled area, to acoustically cryptic non-gas bearing fish southwards towards the Antarctic continent (figure 4c).

## 4. Discussion

Active acoustics can be an invaluable method for monitoring and understanding ecosystems [10]. Since acoustic data are commonly used as a proxy for biomass, a change in the fish community structure, where strong scattering fish are replaced by weak scattering fish, could have considerable implications for ecosystem assessment and modelling of trophic interactions. It has previously been reported that there is a north to south shift in fish community composition in the Scotia Sea [31,32]. This study has confirmed a poleward shift in mesopelagic community structure that parallels a decline in acoustic backscatter. We suggest that the decline is most likely to reflect a shift in the morphological and physiological properties of the fish community present towards the Antarctic continent, rather than a systematic change in total fish biomass.

### (a) Poleward loss of gas-filled swimbladders

The apparent loss of gas-filled swimbladders in fish species with increasing latitude raises interesting questions about

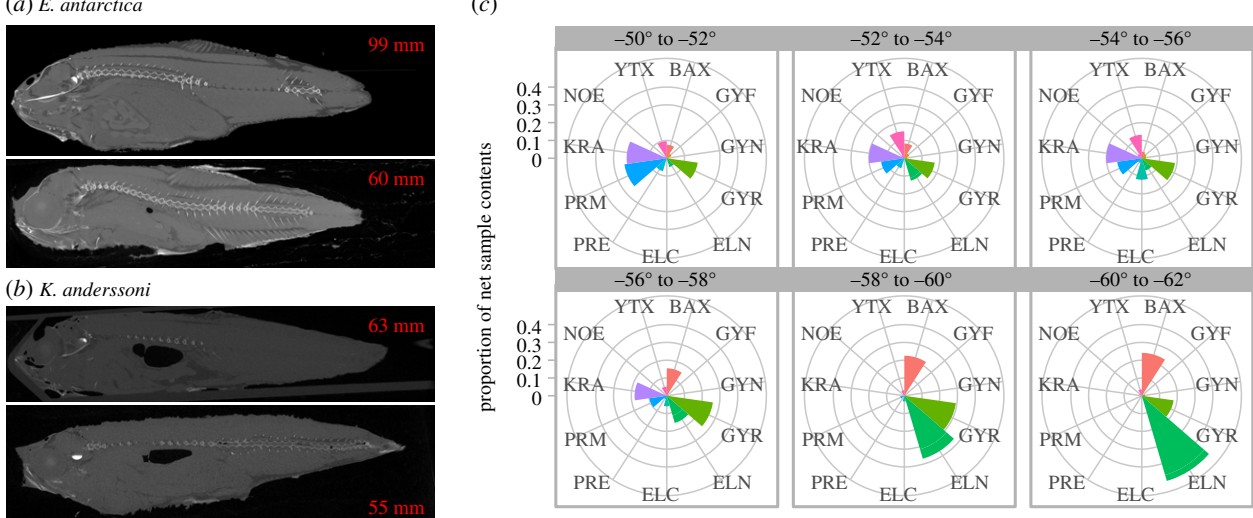

**Figure 3.** Single slice computed tomography scans of (*a*) *Electrona antarctica* showing loss of swimbladder gas and (*b*) *Krefftichthys anderssoni* showing gas presence (dark regions in tissue). Fish standard lengths shown in mm. (*c*) Polar plots of standardized proportions of species captured in 2° latitude bins, each colour segment proportionally corresponds to the abundance of individual species. (Online version in colour.)

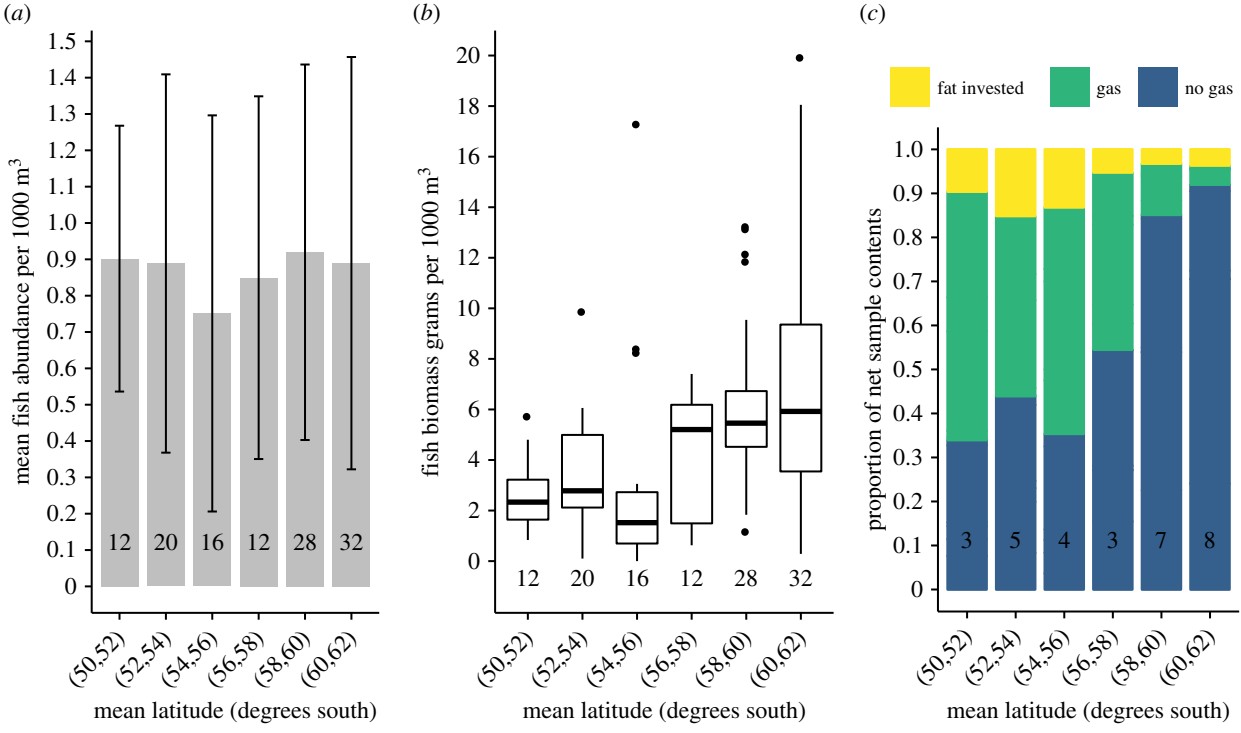

**Figure 4.** (*a*) Mean abundance of fish (individuals per 1000 m$^3$) in RMT25 net samples by latitude. Bars indicate standard deviation between net samples and numbers in columns indicate numbers of individual net strata samples included. (*b*) Biomass of fish (grams per 1000 m$^3$) in RMT25 net samples by latitude, box spans interquartile range (IQR), horizontal line is the median, whiskers include values up to 1.5 × IQR, outlying values plotted individually. (*c*) Relative proportions of fish by swimbladder contents in net samples at latitude. Numbers in columns are the individual number of total water column samples (each consisting of 4 depth strata) used in analysis. (Online version in colour.)

the ecology of the system, and the evolutionary drivers of shifts in swimbladder properties. Typically, mesopelagic fishes undertake large-scale DVM (mean approx. 590 m per cycle) [33], to enable them to forage on abundant near-surface zooplankton at night, while avoiding shallow-water predators during daytime [34]. However, at extreme polar latitudes, DVM is apparently reduced relative to lower latitude habitats [35]. A key underlying factor could be a poleward shift in the light environment, which is known to be an important stimulus of DVM behaviour [36]. It is

therefore plausible that the observed shift in swimbladder morphology is associated with a change in physiological requirements to enable large-scale diurnal depth changes. Species occupying higher latitudes may have a reduced need to alter buoyancy dynamically using a gas-filled swimbladder, instead relying upon buoyancy provided by lipids and avoiding the physiological costs of rapid secretion and resorption of gas. Testing this hypothesis would require modelling of energetic costs of DVM using alternative gas and lipid buoyancy strategies across the depth ranges,

temperatures, water densities and behaviours where diurnal migration takes place in the Southern Ocean [37].

## (b) Ontogenetic shifts in distribution and swimbladder morphology

Data on the presence or absence of gas in swimbladders were restricted to larger size classes of myctophids captured, because small (less than 40 mm) individuals of most species are rarely taken in the Scotia Sea. Saunders *et al.* [38] discussed the absence of larval myctophids in wider Scotia Sea net samples and suggested that many myctophid species of the Scotia Sea could be expatriates from sub-Antarctic, or temperate latitudes that migrate southwards during ontogeny, possibly in search of food hotspots. The main exceptions are *K. anderssoni*, which appears to produce larvae in the coastal waters around South Georgia (Cumberland Bay) [39], and *E. antarctica*, the larvae of which are present in waters towards the Antarctic continental shelf in other regions of the Southern Ocean (Indian Ocean sector) [40]. Whether expatriated myctophids return to waters further north to reproduce remains unclear and requires further investigation.

Unlike the other Southern Ocean myctophids, *E. antarctica* is regarded as a polar specialist that is confined to waters south of the Antarctic Polar Front. This species appears to have a close association with sea ice in some regions of the Southern Ocean (Indian Ocean sector), with the marginal sea ice zone seemingly important for larval development [40]. At present, it is unclear if an ontogenetic habitat shift from sea ice margin to open ocean of *E. antarctica* has favoured the loss of gas-filled swimbladders with increasing body size, but it is plausible that loss of gas represents an adaptation to changing habitat occupancy and DVM behaviour during ontogeny. The observed ontogenetic shift could have importance for interpretation of acoustic data, as any seasonal increase in larval *E. antarctica* with small gas-bearing swimbladders could lead to increased resonance on echograms. Further sampling of smaller individuals of the species in this assemblage, coupled with analyses of their morphology and buoyancy strategies, would clarify if the ontogenetic regression of the swimbladder we observed in *E. antarctica* is unique to that species, or instead more widespread across myctophid species of the region. In particular, further study should examine abundant *Gymnoscopelus* species as we could not rule out gas presence in earlier life stages. It would be advisable to chemically fix larvae and juveniles immediately on capture for later staining and CT scanning, as freezing of such small specimens can lead to tissue damage.

## (c) Challenges for acoustic studies of mesopelagic fish

As in other large-scale surveys of mesopelagic fish biomass [2], we used 38 kHz acoustic data as it generally has sufficient depth resolution to sample the mesopelagic zone. However, the Scotia Sea supports a diverse community of mesopelagic species [41] and single frequency acoustic data lack the detailed information to distinguish between taxa, presenting two main sources of bias. First, fluid-like Antarctic krill *Euphausia superba* would be undetectable individually, but collectively the extensive dense aggregations would be readily detected by echosounders. Second, colonial siphonophores, many species of which bear a gas-filled pneumatophore,

have been shown to be strong acoustic targets with the potential to resonate [3,42,43]. Of 18 siphonophore species known to occur south of 50° S only five are physonect (gas bearing) [44]. While only limited data exist on the abundance of siphonophores in the region, there is evidence that both siphonophores and krill are more prevalent in the south of the Scotia Sea [45–47]. Thus, it seems unlikely that the pattern of a southward reduction in NASC in this study is driven by shifts in the abundance of either krill or physonect siphonophores, but there is a clear need for focused research on the distribution and abundance patterns of siphonophores in the Southern Ocean [3].

Our study shows that reliable interpretation of acoustic biomass survey data requires additional biological information that can be derived by net sampling [7]. Ideally, net sampling and acoustic data collection would occur concurrently. However, limited ship time requires that a balance is achieved between obtaining consistent acoustic transects and acquiring sufficient net data. While much of the acoustic and net sample data used in this study are from longitudinally offset locations and a relatively small regional scale, both datasets span the same major Southern Ocean fronts and water masses (shown in figure 1). This study reveals latitudinal trends in both the acoustics and community structure, which are consistent with other Southern Ocean regions [23]. From an ecological perspective, this is unsurprising as the most common mesopelagic fish typically have circumpolar distributions [12], resulting in broadly analogous latitudinal water masses and habitats [24]. We therefore suggest that the trends revealed in this study may be broadly applicable to the wider Southern Ocean ecosystem. Further net sample and acoustic data would enable tests of the generality of our findings, particularly in the South West Atlantic, South Indian Ocean and South West Pacific Sectors.

It has been noted that there is a markedly greater acoustic backscatter in low latitude mesopelagic habitats relative to those at higher latitudes [22]. A comparison between the Southern Ocean and what are known to be highly productive low latitude sub-tropical regions was not the focus of the current study. Nevertheless, it would be interesting to determine how the morphology of species contrasts between these latitudinal realms, and if fish scattering properties more generally are able to influence patterns of acoustic backscatter across larger global spatial scales.

## (d) Implications for monitoring and modelling

Recent modelling based on acoustic data predicts an increase in mesopelagic biomass under future warming scenarios [22]. Our results indicate that a proportion of the Southern Ocean mesopelagic community is dominated by acoustically cryptic species and therefore polar biomass may be underestimated. It is therefore important that complementary methods of accounting for potential 'missing' biomass are employed, including ground-truthing through net validation. However, such net sampling requires extensive investment in sampling resources, and would be challenging for larger basin- and global-scale surveys [11]. It is possible that the need for such extensive surveys could be partially mitigated by knowledge of basin-scale trends in community composition, as well as backscatter properties of species present, that would enable the development of geographical correction factors that can be applied to acoustics-based estimates. Future solutions may

also lie in the development and refinement of environmental DNA techniques, where acoustic data may be validated and adjusted for through assessment of community composition within water samples [48]. In the meantime, active acoustics in combination with net sampling will remain a powerful combination of methods for the collection of temporal and spatial data for assessment of mesopelagic communities.

# 5. Conclusion

There has been recent interest in the potential exploitation of abundant mesopelagic fish to meet growing human needs, but to achieve this sustainably requires a solid understanding of the impacts on the wider ecosystem [49]. An inability to detect key species during acoustic monitoring presents a particular risk to fish stocks, where species could be exploited beyond sustainable levels. In addition, many fish species have shifted poleward to maintain their optimum thermal tolerance [50–52] as sea temperatures warm, and further shifts are projected. Development of reliable sampling methods, including acoustics, can only enhance our ability to monitor changes in population dynamics of myctophids, informing long-term management of the wider Antarctic ecosystem.

Ethics. All work was completed in compliance with British Antarctic Survey procedures, and the Environmental Protocol (1991) of the Antarctic Treaty. Environmental Impact Assessment numbers: JR161: PA Reference No. PA/2005/05; JR177: Reference No. PA/SG/2007/03; and JR200: PA Reference No. SG2/2008. Special Activity Permits: JR15004: S7 13/2015; JR16003: S7 12/2016. Restricted Activity Permits: JR15002: RAP 2015/006; and JR16003: RAP 2016/032.

Data accessibility. All data are publicly available from the UK Polar Data Centre https://doi.org/10/c4pg, and as part of the electronic supplementary material.

Competing interests. We declare we have no competing interests.

Funding. The scientific cruises (the Polar Ocean Ecosystem Time Series—Western Core Box), R.A.S. and S.F. are funded as part of the Ecosystems Programme at the British Antarctic Survey, Natural Environment Research Council, a part of UK Research and Innovation. T.D. is supported by a NERC GW4+ Doctoral Training Partnership studentship from the Natural Environment Research Council award NE/L002434/1. Stewardship of the acoustic data received support from the European H2020 International Cooperation project MESOPP (Mesopelagic Southern Ocean Prey and Predators, www.mesopp.eu).

Acknowledgements. We thank the crew and scientists of the RRS James Clark Ross for support in sample and acoustic data collection. Tom Davies (University of Bristol) and Ket Smithson (University of Cambridge) supported CT scanning. John Horne provided guidance and practical support with on-board X-ray capture.

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
