## [Reviewer comments · Proceedings of the Royal Society B: Biological Sciences]

Review History

RSPB-2019-0353.R0 (Original submission)

Review form: Reviewer 1

Recommendation

Major revision is needed (please make suggestions in comments)

Scientific importance: Is the manuscript an original and important contribution to its field?

Good

General interest: Is the paper of sufficient general interest?

Good

Quality of the paper: Is the overall quality of the paper suitable?

Acceptable

Is the length of the paper justified?

Yes

Should the paper be seen by a specialist statistical reviewer?

No

Do you have any concerns about statistical analyses in this paper? If so, please specify them explicitly in your report.

No

It is a condition of publication that authors make their supporting data, code and materials available - either as supplementary material or hosted in an external repository. Please rate, if applicable, the supporting data on the following criteria.

Is it accessible?

N/A

Is it clear?

N/A

Is it adequate?

N/A

Do you have any ethical concerns with this paper?

No

Comments to the Author

This is a well written and interesting paper. The introduction provides a good background and motivation for the study, the research question is important and the methods, which include acoustic transects, net sampling and x-ray of fishes are appropriate. The key point is that the proportion of mesopelagic fish with swim bladder decreases southward in the Southern Ocean, which will cause reduced acoustic backscatter relative to any decrease in biomass.

I have, however, one main problem with the manuscript. The authors make a generalization of results made from a very limited part of the Southern Ocean. The acoustic transects presented in this study are all from the most productive part of the Ocean, and all from between 50 and ~60 °S (Fig 1). In arguing their case, there is a sloppy use of references, i.e. in lines 78 – 80, the authors use a study from subtropical waters (20 °S) to 50 °S to state that there is a ubiquitous decline of acoustic backscatter towards the Antarctic landmass in the South Atlantic. That being said, I don't doubt the claim of reduced Antarctic backscatter. The study from the South Pacific sector reaches 70 °S and is highly relevant.

Is there reason to believe that the acoustic properties of Antarctic fish may cause underestimation of biomass?

Yes, the current contribution makes a valid point and the results presented in Figure 4 b are telling. However, when assessing the significant declines in NASC towards increasing latitudes, the authors refer to total biomass when concluding that «there was no decrease in total faunal biomass towards the pole (electronic supplementary material figure S2)». I find this argument misleading. Since this is a paper on fish, the analysis on biomass should rather present how biomass of fish changed along the transect. The invertebrates may be captured by the trawl and contribute to biomass, but not be strong acoustic targets at 38 Khz.

Is there reason to believe that there indeed is a decline in abundance of mesopelagic fish in the Southern Ocean?

Yes; a) these are generally oligotrophic waters. b) Saunders et al. have recently published that the majority of mesopelagic fish in the Southern Ocean are expatriates, i.e. low abundance, (and would give a bias of adult individuals without swim bladders). c) The authors refer to the poleward shift in light environment (photoperiod). Already Dietz 1962 (Scientific American) suggested that vertically migrating mesopelagic fish might have problem coping with the short summer nights near Antarctica. Adding to this, dark waters beneath snow covered ice may hamper visual predation in winter and spring. d) The catches presented in Fig. 4 indeed suggest low abundance (nocturnal catches always < 1 ind 1000 m⁻³).

Therefore, there are both pros and cons to the author's conclusion. The main problem with the current manuscript appears with the authors formulating a general statement based on results from a very limited portion of the Southern Ocean and present their conclusion without caveats. Addressing this point would make this paper a more realistic and valuable contribution.

Review form: Reviewer 2 (Dr Anton Van de Putte)

Recommendation

Accept with minor revision (please list in comments)

Scientific importance: Is the manuscript an original and important contribution to its field?

Excellent

General interest: Is the paper of sufficient general interest?

Excellent

Quality of the paper: Is the overall quality of the paper suitable?

Good

Is the length of the paper justified?

Yes

Do you think some of the material would be more appropriate as an electronic appendix?

No

Should the paper be seen by a specialist statistical reviewer?

No

Do you have any concerns about statistical analyses in this paper? If so, please specify them explicitly in your report.

No

It is a condition of publication that authors make their supporting data, code and materials available - either as supplementary material or hosted in an external repository. Please rate, if applicable, the supporting data on the following criteria.

Is it accessible?

Yes

Is it clear?

Yes

Is it adequate?

N/A

Do you have any ethical concerns with this paper?

No

Comments to the Author

Dear editor and Authors

"Swimbladder morphology masks Southern Ocean mesopelagic fish biomass" studies the morphology of Southern Ocean mesopelagic swims bladder and how this affect their detectability in acoustic surveys. The information presented in this paper is very relevant for accurate estimates of Mesopelagic fish biomass in the Southern Ocean. The paper is well written and clear. And understandable for no-acousticians.

While myctophids are very abundant it seems various factors limit their detectability and as result lead to an underestimate of their abundance and ecological significance. This paper hence is relevant not only for ichthyologist, Southern ocean ecologist but marine biologists in general.

As such I believe it warrants acceptance with minor corrections.

The biggest possible criticisms I can see is regarding the methodology: namely that

1) the fish were not sampled along the acoustic transect hence this might raise concerns that the nets don't directly match the acoustic observation. While I understand the approach (benefits of having a continuous transect vs going back and doing target hauls and messing up data) it would be good to clarify this a bit more in the text.

2) Equally one could raise the argument besides looking at the latitudinal changes in community they should also look at longitudinal changes. Again I understand why this approach was taken an it makes sense but it might be good to clarify that a bit.

To be clear I'm not asking for additional analysis, I would just suggest some clarification on why this approach was chose or others not.

A very trivial remark regarding the sentence below:

"In the Southern Ocean, 33 species of myctophids are present where they form a key component of the Antarctic ecosystem, acting as both predators of zooplankton [12-14]"

It would be good to check the estimated number of species present in the southern Ocean. As part of this i recommend that the authors could provide their definition of the Southern Ocean (I know there are many and none official).

Looking at the references these cite 40 (endemic!), 35, and 35 myctophid species in the Southern Ocean respectively. Also 2 of them Cite Reference 18 from this article So it might begged to include that one here.

Data access

All of the essential swim bladder analysis data is supplied in the supplementary materials, While the station are provided and latitudinal and longitudinal positions can be inferred I would recommend adding the Latitudes and longitudes to Table S1.

I believe most of the additional data of used in this paper including net catches and acoustic

transect are publicly available in various repositories. it would be good to provide links to these repositories.

Best regards
Dr Anton Van de Putte

Decision letter (RSPB-2019-0353.R0)

29-Mar-2019

Dear Ms Dornan:

Your manuscript has now been peer reviewed and the reviews have been assessed by an Associate Editor. The reviewers' comments (not including confidential comments to the Editor) and the comments from the Associate Editor are included at the end of this email for your reference. As you will see, the reviewers and the Editors have raised some concerns with your manuscript. As someone who works on Southern Ocean predators I agree that your work is important and relevant. However, the referees bring up some important points that really must be addressed, because if published your study is likely to have a significant impact on how we study the Southern Ocean. We would thus like to invite you to revise your manuscript to address them.

Research ethics:

Use of animals and field studies:

Please submit a copy of your revised paper within three weeks. If we do not hear from you within this time your manuscript will be rejected. If you are unable to meet this deadline please let us know as soon as possible, as we may be able to grant a short extension.

Best wishes,

Proceedings B
mailto: proceedingsb@royalsociety.org

Associate Editor

Board Member: 1

Comments to Author:

Both reviewers are supportive of publication of the manuscript, however have asked the authors for clarification around a number of components of the study, particularly in reference to mismatches between the trawl and acoustic transects (and therefore lack of direct comparison). Please ensure that you directly address each of the reviewers comments and clearly identify how the manuscript has been revised in response.

Reviewer(s)' Comments to Author:

Referee: 1

Comments to the Author(s)

This is a well written and interesting paper. The introduction provides a good background and motivation for the study, the research question is important and the methods, which include acoustic transects, net sampling and x-ray of fishes are appropriate. The key point is that the proportion of mesopelagic fish with swim bladder decreases southward in the Southern Ocean, which will cause reduced acoustic backscatter relative to any decrease in biomass.

I have, however, one main problem with the manuscript. The authors make a generalization of results made from a very limited part of the Southern Ocean. The acoustic transects presented in this study are all from the most productive part of the Ocean, and all from between 50 and ~60 °S (Fig 1). In arguing their case, there is a sloppy use of references, i.e. in lines 78 - 80, the authors use a study from subtropical waters (20 °S) to 50 °S to state that there is a ubiquitous decline of acoustic backscatter towards the Antarctic landmass in the South Atlantic. That being said, I don't doubt the claim of reduced Antarctic backscatter. The study from the South Pacific sector reaches 70 °S and is highly relevant.

Is there reason to believe that the acoustic properties of Antarctic fish may cause underestimation of biomass?

Yes, the current contribution makes a valid point and the results presented in Figure 4 b are telling. However, when assessing the significant declines in NASC towards increasing latitudes, the authors refer to total biomass when concluding that «there was no decrease in total faunal biomass towards the pole (electronic supplementary material figure S2)». I find this argument misleading. Since this is a paper on fish, the analysis on biomass should rather present how biomass of fish changed along the transect. The invertebrates may be captured by the trawl and contribute to biomass, but not be strong acoustic targets at 38 KHz.

Is there reason to believe that there indeed is a decline in abundance of mesopelagic fish in the Southern Ocean?

Yes; a) these are generally oligotrophic waters. b) Saunders et al. have recently published that the majority of mesopelagic fish in the Southern Ocean are expatriates, i.e. low abundance, (and would give a bias of adult individuals without swim bladders). c) The authors refer to the poleward shift in light environment (photoperiod). Already Dietz 1962 (Scientific American) suggested that vertically migrating mesopelagic fish might have problem coping with the short summer nights near Antarctica. Adding to this, dark waters beneath snow covered ice may hamper visual predation in winter and spring. d) The catches presented in Fig. 4 indeed suggest low abundance (nocturnal catches always < 1 ind 1000 m-3).

Therefore, there are both pros and cons to the author's conclusion. The main problem with the current manuscript appears with the authors formulating a general statement based on results from a very limited portion of the Southern Ocean and present their conclusion without caveats. Addressing this point would make this paper a more realistic and valuable contribution.

Referee: 2

Comments to the Author(s)

Dear editor and Authors

"Swimbladder morphology masks Southern Ocean mesopelagic fish biomass" studies the morphology of Southern Ocean mesopelagic swims bladder and how this affect their detectability in acoustic surveys. The information presented in this paper is very relevant for accurate estimates of Mesopelagic fish biomass in the Southern Ocean. The paper is well written and clear. And understandable for no-acousticians.

While myctophids are very abundant it seems various factors limit their detectability and as result lead to an underestimate of their abundance and ecological significance. This paper hence is relevant not only for ichthyologist, Southern ocean ecologist but marine biologists in general.

As such I believe it warrants acceptance with minor corrections.

The biggest possible criticisms I can see is regarding the methodology: namely that

1) the fish were not sampled along the acoustic transect hence this might raise concerns that the nets don't directly match the acoustic observation. While I understand the approach (benefits of having a continuous transect vs going back and doing target hauls and messing up data) it would be good to clarify this a bit more in the text.

2) Equally one could raise the argument besides looking at the latitudinal changes in community they should also look at longitudinal changes. Again I understand why this approach was taken an it makes sense but it might be good to clarify that a bit.

To be clear I'm not asking for additional analysis, I would just suggest some clarification on why this approach was chose or others not.

A very trivial remark regarding the sentence below:

"In the Southern Ocean, 33 species of myctophids are present where they form a key component of the Antarctic ecosystem, acting as both predators of zooplankton [12-14]"

It would be good to check the estimated number of species present in the southern Ocean. As part of this i recommend that the authors could provide their definition of the Southern Ocean (I know there are many and none official).

Looking at the references these cite 40 (endemic!), 35, and 35 myctophid species in the Southern Ocean respectively. Also 2 of them Cite Reference 18 from this article So it might begged to include that one here.

Data access

All of the essential swim bladder analysis data is supplied in the supplementary materials, While the station are provided and latitudinal and longitudinal positions can be inferred I would recommend adding the Latitudes and longitudes to Table S1.

I believe most of the additional data of used in this paper including net catches and acoustic transect are publicly available in various repositories. it would be good to provide links to these repositories.

Best regards

Dr Anton Van de Putte

Author's Response to Decision Letter for (RSPB-2019-0353.R0)

See Appendix A.

Decision letter (RSPB-2019-0353.R1)

26-Apr-2019

Dear Ms Dornan

I am pleased to inform you that your manuscript RSPB-2019-0353.R1 entitled "Swimbladder morphology masks Southern Ocean mesopelagic fish biomass" has been accepted for publication in Proceedings B.

The referee(s) have recommended publication, but also suggest some minor revisions to your manuscript. Therefore, I invite you to respond to the referee(s)' comments and revise your manuscript. Because the schedule for publication is very tight, it is a condition of publication that you submit the revised version of your manuscript within 7 days. If you do not think you will be able to meet this date please let us know.

[http://datadryad.org/submit?journalID=RSPB&manu=\(Document not available\)](http://datadryad.org/submit?journalID=RSPB&manu=(Document%20not%20available)) which will take you to your unique entry in the Dryad repository. If you have already submitted your data to dryad you can make any necessary revisions to your dataset by following the above link. Please see <https://royalsociety.org/journals/ethics-policies/data-sharing-mining/> for more details.

Sincerely,

Proceedings B
mailto:proceedingsb@royalsociety.org

Associate Editor:

Comments to Author:

The authors have undertaken to address all of the reviewers comments and as a result I am happy to progress this to publication. There are however, a number of grammatical corrections and clarification of text needed to improve the readability of the manuscript. I have identified these in the attached version of the manuscript.

Author's Response to Decision Letter for (RSPB-2019-0353.R1)

See Appendix B.

Decision letter (RSPB-2019-0353.R2)

02-May-2019

Dear Ms Dornan

I am pleased to inform you that your manuscript entitled "Swimbladder morphology masks Southern Ocean mesopelagic fish biomass" has been accepted for publication in Proceedings B.

Your article has been estimated as being 9 pages long. Our Production Office will be able to confirm the exact length at proof stage.

Open Access

You are invited to opt for Open Access, making your freely available to all as soon as it is ready for publication under a CCBY licence. Our article processing charge for Open Access is £1700. Corresponding authors from member institutions

Paper charges

Sincerely,

Proceedings B
mailto: proceedingsb@royalsociety.org

Appendix A

Response to reviewers

(Please note that all line numbers relate to track change full mark up.)

Reviewer 1

Comment	Response
1. The authors make a generalization of results made from a very limited part of the Southern Ocean. The acoustic transects presented in this study are all from the most productive part of the Ocean, and all from between 50 and ~60 °S (Fig 1).	We acknowledge the oversight in explaining our reasoning for wider extrapolation beyond localised survey data. We consider the extrapolation to the Southern Ocean valid as the species we capture in the Scotia Sea generally have an Antarctic circumpolar distribution (Hulley, 1990). We now make it clear that the findings have broader relevance by highlighting that barriers to species distributions are latitudinal because of oceanographic conditions, rather than longitudinal (lines 311-325).
2. In arguing their case, there is a sloppy use of references, i.e. in lines 78 – 80, the authors use a study from subtropical waters (20 °S) to 50 °S to state that there is a ubiquitous decline of acoustic backscatter towards the Antarctic landmass in the South Atlantic. That being said, I don't doubt the claim of reduced Antarctic backscatter. The study from the South Pacific sector reaches 70 °S and is highly relevant.	Whilst the reference from subtropical data does not extend to the Antarctic continent, it does cross into Antarctic polar frontal water masses. As there is very limited available data in the Indian Ocean sector of the Southern Ocean, it was included for completeness. That said, we do agree that the inclusion of this reference may be misleading and have removed it and amended the context of the sentence (lines 79-81 and line74).
3. Is there reason to believe that the acoustic properties of Antarctic fish may cause underestimation of biomass? Yes, the current contribution makes a valid point and the results presented in Figure 4 b are telling. However, when assessing the significant declines in NASC towards increasing latitudes, the authors refer to total biomass when concluding that «there was no decrease in total faunal biomass towards the pole (electronic supplementary material figure S2)». I find this argument misleading. Since this is a paper on fish, the analysis on biomass should rather present how biomass of fish changed along the transect. The invertebrates may be captured by the trawl and contribute to biomass, but not be strong acoustic targets at 38 kHz.	We presented our findings in terms of fish abundance yet were highlighting our conclusion in terms of biomass. We have now analysed fish catch biomass data from the same net samples used in the abundance analysis, and now present these biomass data in figure 4b (catch data source added to methods, line 127-128 and 174-178). Our overall findings remain unchanged (referred to on lines 224-226). We have kept the supplementary figure S2, since large aggregations of non-fish fauna such as Antarctic krill (Euphausia superba) can contribute to acoustic backscatter albeit to a lesser extent at lower frequencies. Addressing this was our main concern when assessing if acoustic backscatter was being biased by non-fish species exhibiting a general decline in biomass in the supplementary material. To fully address the reviewers' comments we have added supplementary figure S3 to show the same data for fish biomass only. We refer to this on lines 189-193.

4. Is there reason to believe that there indeed is a decline in abundance of mesopelagic fish in the Southern Ocean? Yes; a) these are generally oligotrophic waters. b) Saunders et al. have recently published that the majority of mesopelagic fish in the Southern Ocean are expatriates, i.e. low abundance, (and would give a bias of adult individuals without swim bladders). c) The authors refer to the poleward shift in light environment (photoperiod). Already Dietz 1962 (Scientific American) suggested that vertically migrating mesopelagic fish might have problem coping with the short summer nights near Antarctica. Adding to this, dark waters beneath snow covered ice may hamper visual predation in winter and spring. d) The catches presented in Fig. 4 indeed suggest low abundance (nocturnal catches always < 1 ind 1000 m-3).	Our results show that within the bounds of the Southern Ocean (defined in lines 78-79) the total biomass of mesopelagic fish does not decline towards the poles, whilst acoustic estimates do indicate a decline. To address the specific question of whether there is a smaller biomass of fish in the Southern Ocean compared with elsewhere requires a greater geographic context as the reviewer identifies, and this point is now highlighted in lines 327-333. Whilst the reviewer raises many reasons why there may be less biomass in the Southern Ocean than further north, at present, we do not have the acoustic or net data to explore these fully as this issue was not the focus of the current study. We are keen to reiterate the key message of this work is that the use of active acoustics as a means of assessing mesopelagic fish abundance in the Southern Ocean is likely to underestimate the biomass of fish as we move towards the continent due to a clear morphological change in the community from sub Antarctic to polar waters.
5. The main problem with the current manuscript appears with the authors formulating a general statement based on results from a very limited portion of the Southern Ocean and present their conclusion without caveats. Addressing this point would make this paper a more realistic and valuable contribution.	We have stated our definition of the Southern Ocean on lines 77-78, we have removed the mis-leading reference and we have introduced a discussion regarding latitudinal boundaries and lack of longitudinal boundaries to explain the context of our statements. We feel this has incorporated the caveats required by the reviewer.

Reviewer 2

1) the fish were not sampled along the acoustic transect hence this might raise concerns that the nets don't directly match the acoustic observation. While I understand the approach (benefits of having a continuous transect vs going back and doing target hauls and messing up data) it would be good to clarify this a bit more in the text.	The reviewer acknowledges the constraints between sampling pressures and we have highlighted the geographic locations of both our net and acoustic sampling in figure 1. During our discussion regarding latitudinal and longitudinal barriers to mesopelagic fish species distribution we have further clarified why we (and the reviewer) believe this approach is appropriate (lines 311-325).
---	--

2) Equally one could raise the argument besides looking at the latitudinal changes in community they should also look at longitudinal changes. Again I understand why this approach was taken and it makes sense but it might be good to clarify that a bit.	This was addressed in response to previous reviewers comment regarding barriers to species distributions are generally latitudinal because of oceanographic conditions, rather than longitudinal. The map does have circumpolar oceanographic fronts on it, showing oceanographic changes generally occur in the latitudinal realm. We draw this to the readers attention in line 317-321.
3) A very trivial remark regarding the sentence below: "In the Southern Ocean, 33 species of myctophids are present where they form a key component of the Antarctic ecosystem, acting as both predators of zooplankton [12-14]" It would be good to check the estimated number of species present in the southern Ocean. As part of this i recommend that the authors could provide their definition of the Southern Ocean (I know there are many and none official). Looking at the references these cite 40 (endemic!), 35, and 35 myctophid species in the Southern Ocean respectively. Also 2 of them Cite Reference 18 from this article So it might begged to include that one here.	We have corrected the text to read 35 species, and relocated the relevant citation from the end of the sentence to the appropriate location (Hulley, 1990, [18] line 65, moved to [12] line 62) as recommended by reviewer. Subsequent citation numbers have been updated as required. We have also defined the Southern Ocean as a region south of 50° S for the purposes of this study (lines 77-78).
4) Data access All of the essential swim bladder analysis data is supplied in the supplementary materials, While the station are provided and latitudinal and longitudinal positions can be inferred I would recommend adding the Latitudes and longitudes to Table S1.	Latitudes and longitudes have been added to Table S1 as requested.
5) I believe most of the additional data of used in this paper including net catches and acoustic transect are publicly available in various repositories. It would be good to provide links to these repositories.	All data is available on UK Polar data centre and links are now included. Whilst currently under embargo these will be freely available on publication of manuscript. DOI link: https://doi.org/10.5285/e15d622c-5c7e-45e9-b127-f27def94bbe8 Short DOI link: https://doi.org/10/c4pq Short DOI: 10/c4pq

Appendix B

Response to reviewers

(Please note that all line numbers relate to track changes in 'All Markup' view.)

Reviewer 1

Comment	Response
Rephrase sentence for consistency in the use of "person" to : "We also show using catch data from survey trawls that the fish community switches from fish possessing gas-filled swimbladders to those lacking swimbaldders as latitude increases towards the Antarctic continent"	Line 20-23, rephrased as requested.
replace "present" with "known to occur" - this then accounts for the potential that we actually haven't described all species (which in remote regions is highly likely)	Line 64, text replaced as requested.
If you have grouped seals together (knowing that both phocids and otariids occur in the Sthn Ocean and both eat myctophids) why not group seabirds?	Line 66, rephrased to group seabirds.
Rephrase sentence for consistency in the use of "person" to : "We conclude that the reduction in backscatter with latitude towards Antarctica is strongly influenced by a shift in community structure..."	Line 87-88, rephrased as requested.
insert "where data was " so sentence reads "...with the exception of JR161 and JR200 where data was collected to 800 m and 990 m..."	Line 101, text inserted as requested.
Not sure if "bad data regions" is a common term, but sounds odd. Happy to accept if it is but if not suggest rewording the sentence to "Prior to integration, areas where false bottom echoes, the seabed, surface near-field intermittent noise and attenuated signals occurred were set to "no-data"..."	Line 104, replaced "bad data regions, including false bottom echoes..." with "bad or unwanted data such as false bottom echoes..."
Sentences on lines 103-105 need context - why were they excluded	Line 106-109, have added context to explain why data was excluded
Lines 107-109 need some clarification - First, why is a high NASC value a trigger for checking validity? Second what was removed from analysis is not clear - the less than 10% of the 1% visually scrutinised or all 1%?	Line 111-113, have highlighted why high NASC values were scrutinised and clarified data removed.
Insert "the" so the sentence reads "...spanning the frontal regions and water masses..."	Line 121, text inserted as requested.
Remove "however"	Line 122, text removed as requested.
Already identified on line 115 that the sample was stratified and then detail that stratification on line 123 so no need to repeat here	Line 126, deleted "for stratified sampling" as requested.

Insert "The" so reads "The total catch weight..."	Line 132, text inserted as requested.
insert "and the" so reads "... and the standard length measured before..."	Line 134, text inserted as requested.
clarification needed: the same cruises or other cruises?	Line 139, these were the same cruises, we have amended text to state this explicitly.
Clarification needed: is this 7 of some unknown number of the most common species or the seven most common species?	Line 142-143, reworded to seven of the eight most common, based on the net data.
Some clarifying words or a sentence is needed here given that swimbladder assessments were carried out on 7 species, while community composition analysis was carried out on 11 species - presumably swimbladder status could not be assigned to all 11 species included in the community composition analyses	Line 190-191, we have clarified that four species could be assigned from the literature, seven species were looked at here.
replace "towards" with "with"	Line 196, text replaced as requested.
Clarification in regards to what trend is being referred to here is needed - the greatest deviation from the overall trend?	Line 197-198, this has been amended to more correctly read "variability" and trend is explicitly defined
for consistency in terminology this would be better phrased as "...there was no decrease in biomass with increasing latitude..."	Line 204, rephrased as requested.
this would be better phrased as "across all lengths assessed". At the moment it could be interpreted that those animals that were not small of large (i.e. medium) were doing something different.	Line 209-210, rephrased as requested.
Rework sentence to "Gymnoscopelus braueri (SL 68 mm-123 mm, n=21), G. nicholsi (SL 124 mm-153 mm, n=14) and G. fraseri (SL 55 mm-84 mm, n=12) showed no evidence of swimbladder gas."	Line 211-213, rephrased as requested.
ontogenetic implies from juvenile to adult so no need to repeat	Line 214, deleted "from juvenile to adults" to remove repetition.
for consistency replace "towards the pole" with "with increasing latitude"	Line 232, text replaced as requested.
what is this regressed referring to? It doesn't particularly make sense as it is currently presented in the sentence	Line 233-234, rephrased to "regressed and non-gas-bearing swimbladder" for clarification
This sentence and the next should be moved to the "Gas presence and absence" section	Line 240-246, a modified version of sentence has been retained in the current paragraph with minor corrections to text as it provides an overview of the categorisations used in the changing community morphology as presented in figure 3c.
This should be in the methods: "Species not assessed for gas component as part of this study..."	Line 243-248, moved to methods as suggested (lines 172-177).

Rephrase this sentence to " The apparent loss of gas filled swimbladders in fish species with increasing latitude raises interesting..."	Line 266, rephrased as requested.
replace "that has larvae present" with "the larvae of which are present"	Line 291, rephrased as requested.
Not clear what the ontogenetic habitat shift is if the species is associated with sea ice and larvae are also associated with sea ice - some clarification is needed.	Line 299-301, we have reworded this sentence to clarify the meaning of our statement on ontogenetic habitat use.
some context as to why this genus has been identified as 'in particular' (and importantly why none of the others without gas) is needed.	Line 308-309, we have added requested context. Gymnoscopelus might possess gas in smaller individuals than were present in our catches (line 308-309). Evidence suggests that the two other 'non-gas' species (Bathylagus and Notolepis) do not possess evidence of swimbladders at any stage, added to line 174-175.